# Elderberry (*Sambucus nigra* L.) Encapsulated Extracts as Meat Extenders against Lipid and Protein Oxidation during the Shelf-Life of Beef Burgers

**DOI:** 10.3390/antiox11112130

**Published:** 2022-10-28

**Authors:** Gabriele Rocchetti, Pier Paolo Becchi, Luigi Lucini, Aurora Cittadini, Paulo E. S. Munekata, Mirian Pateiro, Rubén Domínguez, José M. Lorenzo

**Affiliations:** 1Department of Animal Science, Food and Nutrition, Università Cattolica del Sacro Cuore, Via Emilia Parmense 84, 29122 Piacenza, Italy; 2Department for Sustainable Food Process, Università Cattolica del Sacro Cuore, Via Emilia Parmense 84, 29122 Piacenza, Italy; 3Centro Tecnológico de la Carne de Galicia, Parque Tecnológico de Galicia, Avd. Galicia N° 4, San Cibrao das Viñas, 32900 Ourense, Spain; 4Instituto de Innovación y Sostenibilidad en la Cadena Agroalimentaria (IS-FOOD), Universidad Pública de Navarra (UPNA), Arrosadia Campus, 31006 Pamplona, Spain; 5Área de Tecnología de los Alimentos, Facultad de Ciencias de Ourense, Universidad de Vigo, 32004 Ourense, Spain

**Keywords:** polyphenols, natural additives, foodomics, oxidative degradation, storage, healthy meat, functional foods

## Abstract

In this work, we studied the impact of encapsulated elderberry extracts as natural meat extenders to preserve both the quality and the oxidative stability of beef burgers. In particular, the comprehensive chemical changes of beef burgers treated with different antioxidants, namely, (a) a control without antioxidants, (b) 0.5 g/kg sodium erythorbate (ERY), (c) 2.5 g/kg encapsulated elderberry extract (EE 2.5), and (d) 5 g/kg encapsulated elderberry extract (EE 5), each one packaged under modified atmosphere (80% O_2_ and 20% CO_2_) for 13 days storage at 2 ± 1 °C, were deeply evaluated. Overall, EEs showed a wide array of antioxidant compounds, namely polyphenols like anthocyanins, flavonols, and phenolic acids. Multivariate statistics provided marked chemical differences between burgers manufactured with EEs and synthetic antioxidants (ERY) during 13-days storage in terms of both metabolomic profiles and typical lipid/protein oxidation markers (such as malondialdehyde and total carbonyls). Most of the differences could be attributed to some discriminant compounds, namely glutathione, 4-hydroxy-2-nonenal, hydroxy/peroxy-derivatives of fatty acids, carbonyl compounds (such as 5-nonen-2-one and 1,5-octadien-3-one), and cholesterol. Interestingly, significant correlations (*p* < 0.01) were observed between malondialdehyde, total carbonyls, and these discriminant metabolites. The combination of spectrophotometric approaches and a high-throughput untargeted metabolomics analysis outlined a strong modulation of both lipid and protein oxidations, likely promoted by the encapsulated meat extender (elderberry), thus confirming its ability to delay oxidative phenomena during the shelf-life of beef burgers.

## 1. Introduction

Before being consumed, meat products undergo a series of processes that can impair the quality of the finished products [1]. In this regard, meat products are rich in proteins and, depending on the muscle type, contain variable quantities and proportions of both storage lipids (triacylglycerols) and structural lipids (phospholipids). Such proteins and lipids represent the most susceptible targets for oxidation and degradation processes [2]. In meat, lipids are mainly organized into triglycerides and phospholipids, and the latter are the main contributors to the development of lipid oxidation [3]. Regarding proteins, the complexity of the reactions involved in their oxidation, together with several other factors, make the mechanisms of protein oxidation difficult to fully understand [2].

Oxidative reactions are extremely important in meat science since they generate several toxic and harmful compounds, thus decreasing the nutritional properties and sensory quality of the product during storage. Therefore, several strategies (based on both natural and synthetic strategies) have been inspected in recent years to mitigate these phenomena [4,5]. From this perspective, the application of plant extracts to meat and meat products is an important strategy to increase their shelf-life [6]. Interestingly, among all natural water-soluble pigments, the anthocyanins that are present in several berries are promising additives that possess both important antioxidant activity and intense color [7]. Accordingly, elderberry (*Sambucus nigra* L.) has been described as an important source of water-soluble anthocyanins, also containing some other compounds, such as flavonols and phenolic acids [8]. Indeed, these compounds are well-known free radical scavengers and can protect the human body against oxidative stress and lipid peroxidation processes [9]. Elderberry is a good source of proteins, with content in fruits at 2.8%, in flowers at 2.4%, and in leaves at 3.3% [10], thus representing good candidates as meat extenders and functional ingredients. However, it is important to consider that the content and profile of bioactive compounds of the plant depend on different parameters and differs between wild edible plants and cultivated elderberries or between varieties, location, growing season, ripening stage, and climatic conditions [11]. However, despite the factors previously cited, elderberries possess a high potential since they have a higher anthocyanin content and in vitro antioxidant activity when compared with other important berries, such as blackcurrant and raspberry.

Although originally devised to reduce costs by replacing raw meat with cheaper components, the use of properly selected ingredients from natural origin, also known as extenders, presents great opportunities for incorporating some healthier ingredients and/or bioactive components into the final meat product [12]. Within this scenario, the integration of plant-based ingredients into meat products has already been proven as a successful and consumer-accepted strategy, paving the way to different approaches based on the reformulation of healthier and more sustainable meat products [13]. Therefore, meat extenders from natural origin and plant extracts represent a good opportunity to improve both the chemical quality of the product through its shelf-life period and, at the same time, the sustainability of its production cycle.

Starting from these background conditions, in this study, we evaluated the potential impact of encapsulated elderberry extracts as a natural meat extender to preserve both the quality and stability of beef burgers. To this aim, we coupled classical spectrophotometric assays (providing known oxidation-related markers, such as MDA and total carbonyls) to an untargeted metabolomics-based approach to provide new insights into the modulation of the encapsulated extracts of the meat quality following 13-days of the shelf-life period.

## 2. Materials and Methods

### 2.1. Plant Material and Extract Preparation

This technological development and the product are protected by patent (Application number: P202230560/Date: 23 June 2022/patent pending). The protection of this patent includes (i) all the stages of the production process for obtaining and stabilizing (encapsulation) bioactive compounds from elderberry, (ii) the final product obtained (encapsulated elderberry extract), and (iii) its application in the meat industry. Therefore, the steps described below are included in the patent protection but do not limit its scope (see Section 5).

The elderberries were collected from wild trees and washed, selected, and lyophilized according to the process described by Domínguez et al. [9]. The bioactive compounds were extracted from lyophilized elderberries (solid/solvent ratio 1:20) using ethanolic solution (50%) at pH 2 and 60 °C for 5 h (shaking manually every hour). Once the extraction was complete, the ethanol was evaporated in a rotary evaporator (25 min, 75 °C, and vacuum). The concentrated extract was encapsulated with a spray-dryer (Mini Spray-Dryer B-290; Büchi), using maltodextrin as encapsulating agent (13.5 g/100 g extract). The inlet temperature (spray chamber) was set at 145 °C and the extractor speed at 80%, while the feed flow was set at 600 g/h. The encapsulated elderberry extract (EE) obtained was vacuum packed and stored frozen (−20 °C) until use.

### 2.2. Preparation of Beef Burgers

The present research was performed in the Centro Tecnolóxico da Carne de Galicia (San Cibrao das Viñas, ES, Spain). Four different batches of beef burgers were manufactured: Control batch [CON], without any additives, a positive control [ERY] which includes synthetic antioxidant (sodium erythorbate at 0.5 g/kg) and two additional batches formulated with encapsulated elderberry extract at 2.5 g/kg [EE 2.5] and 5 g/kg [EE 5]. The formulation of all batches consisted of beef meat (91.5%) containing approximately 5% fat, salt (1%), and water (7.5%). This formulation was selected without the addition of any other ingredient (additives, condiments, spices, etc.) in order to observe the potential of the encapsulated extract exclusively, and there were no interferences from other ingredients.

The lean beef meat was ground through an 8 mm diameter plate in a refrigerated mincer machine (La Minerva, Bologna, IT, Italy). In the case of batches with antioxidants (erythorbate or encapsulated elderberry extract), each antioxidant was previously diluted in water, and the mass of each batch (meat, salt, and water) was mixed to achieve a homogeneous distribution of all the ingredients. Finally, burgers were shaped with a manual burger machine, approximately 10 cm in diameter and 1 cm high (100 g per hamburger). After manufacture, beef burgers were packed in MAP conditions (80:20 O_2_:CO_2_) using polystyrene trays, which were sealed with polyethylene film [permeability < 2 mL/(m^2^ bar/day)]. The packaging was carried out using a heat sealer (LARI3/Pn T-VG-R-SKIN, Ca.Ve.Co., Palazzolo sull’Oglio, IT, Italy). In order to simulate supermarket conditions, the packaged samples were stored at 2 ± 1 °C under the light. Beef burgers were sampled after 1 (T0), 5 (T5), and 13 (T13) days of storage. The whole experiment was replicated three times with the same ingredients and formulation. Overall, 144 burger samples (four burgers per batch × four different processing batches × three manufacture replicates × three sampling days) were produced and analyzed. The visual aspect of the burgers formulated can be found in Figure 1.

### 2.3. Lipid and Protein Oxidation Parameters

The lipid oxidation of beef burgers was evaluated with the thiobarbituric acid reactive substances (TBARS) assay [14], expressing the results as mg malondialdehyde per kg of the sample (mg MDA/kg). The protein oxidation measurement was carried out based on the carbonyl derivatization with 2,4-dinitrophenyl hydrazine and their quantification by spectrophotometry (370 nm) using the procedure described by de Carvalho et al. [15]. In this case, the results were expressed as nmol carbonyl/ mg protein.

### 2.4. Untargeted Profiling by HRMS of Beef Burgers during Storage

In this work, 1 g of each sample was extracted in 10 mL of an 80% methanolic solution acidified with 0.1% formic acid. The extraction was promoted by a homogenizer-assisted extractor (Ultra-turrax, Ika T25, Staufen, BW, Germany) [16]. The extracts were centrifuged (6000× *g* for 15 min at 4 °C) and then filtered using 0.22 μm cellulose syringe filters. Finally, the filtered solutions were collected in amber vials for metabolomics-based profiling. The untargeted analysis was based on high-resolution mass spectrometry (HRMS) performed on a Q-Exactive™ Focus Hybrid Quadrupole-Orbitrap Mass Spectrometer (Thermo Scientific, Waltham, MA, USA) coupled to a Vanquish ultra-high-pressure liquid chromatography (UHPLC) pump and equipped with heated electrospray ionization (HESI)-II probe (Thermo Scientific, MA, USA). A gradient elution of water-acetonitrile (both LC-MS grade, from Sigma-Aldrich, Milan, IT, Italy) (6–94% acetonitrile in 35 min) was used for the chromatography step, using 0.1% formic acid as phase modifier on a BEH C18 column (2.1 × 100 mm, 1.7 μm) (Milford, MA, USA). The HRMS conditions were adapted from our previously published work [17]. We used a flow rate equal to 200 μL/min, selecting a full scan MS analysis based on a positive ionization mode and a mass resolution of 70,000 at *m*/*z* 200. The injection volume was 6 μL, with a *m*/*z* range of 80–1200. The automatic gain control target (AGC target) and the maximum injection time (IT) were 1 × 10^6^ and 200 ms, respectively. Also, pooled quality control (QC) samples were randomly acquired in a data-dependent (Top N = 3, using typically normalized collision energies of 10, 20, and 40 eV) MS/MS mode, with full scan mass resolution reduced to 17,500 at *m*/*z* 200, with an AGC target value of 1 × 10^5^, maximum IT of 100 ms, and isolation window of 1.0 *m*/*z*, respectively. The HESI parameters are reported in previous work by our research group [17]. The injection sequence was also randomized to avoid any possible bias. The raw data files were then processed using the software MS-DIAL (version 4.80, RIKEN Center for Sustainable Resource Science: Metabolome Informatics Research Team, 1-7-22 Suehiro-cho, Tsurumi-ku, Yokohama, Kanagawa, Japan.) [18]. The workflow consisted of an automatic peak finding, LOWESS normalization, and annotation via spectral matching (against the databases FooDB). The mass range of 80–1200 *m*/*z* was searched for features with a minimum peak height of 10,000 cps. The MS and MS/MS tolerance for peak centroiding was set to 0.05 and 0.1 Da, respectively. Accurate mass tolerance for identification was 0.05 Da for MS and 0.1 Da for MS/MS. The identification step was based on several criteria, such as mass accuracy, isotopic pattern, and spectral matching. The total identification score cut-off was 50%, considering the most common HESI + adducts. Gap filling using peak finder algorithm was performed to fill in missing peaks, considering 5 ppm tolerance for *m*/*z* values. Therefore, in our experimental condition, a level 2 of confidence in annotation [19] for untargeted metabolomics experiments was achieved. Finally, the total anthocyanin content of the raw elderberry extract used for the trial was determined by UHPLC-HRMS, using a semi-quantitative analysis, and expressing the results as cyanidin Equivalents (g/kg).

### 2.5. Statistical Analysis

A one-way analysis of the variance (ANOVA) was done using the software PASW Statistics 26.0 (SPSS Inc., Chicago, IL, USA) to investigate significant differences (*p* < 0.05, Duncan’s post hoc test) when considering the results of each in vitro assay related to both lipid and protein oxidation. Additionally, Pearson’s correlation coefficients were inspected by means of the software SPSS (version 26.0, SPSS Inc., Chiacgo, IL, USA) to check correlations between malondialdehyde (MDA) and total carbonyl contents with the discriminant metabolites related to oxidation phenomena. The metabolomics data were then elaborated using two different software, namely MetaboAnalyst (version 5.0) [20] and SIMCA (version 13; from Umetrics, Malmo, SE, Sweden). Data processing and normalization were carried out as previously described by our research group [21]. After data normalization of the mass features, both unsupervised and supervised multivariate statistics were carried out based on hierarchical cluster analysis (HCA, Euclidean distance) and orthogonal projections to latent structures discriminant analysis (OPLS-DA). For supervised statistics, we considered (a) the impact of storage time, (b) the combination of storage time × treatment type, and (c) the changes in metabolomic profiles after 13 days of storage. The OPLS-DA model validation parameters (goodness-of-fit R^2^Y together with goodness-of-prediction Q^2^Y) were inspected, considering a Q^2^Y prediction ability of >0.5 as the acceptability threshold. Thereafter, the OPLS-DA model was checked for outliers and validated by CV-ANOVA (*p* < 0.01), and permutation testing (N > 200) was performed to exclude model overfitting. The importance of each metabolite for discrimination purposes was then calculated according to the VIP (i.e., variables importance in projection) selection method, considering as the minimum significant threshold those values higher than 1. Additionally, a Fold-Change analysis (cut-off > 1.2) was done to inspect the up- or down-accumulation of the discriminant metabolites outlined by the VIP approach.

## 3. Results and Discussion

### 3.1. Phytochemical Profiling of the Elderberry Extract

The comprehensive phytochemical profile of the elderberry extract added to beef burgers was investigated through an untargeted metabolomics-based approach. Overall, the UHPLC-HRMS analysis provided a dataset consisting of 379 phenolics, with flavonoids (53%) and phenolic acids (both hydroxycinnamics and hydroxybenzoics; 18%) representing the most abundant classes of compounds. Detailed information regarding the identification of these compounds (according to a level 2 of confidence in annotation, typical of untargeted metabolomics-based workflows) is provided in Appendix A. Interestingly, the injection of pooled QC samples combined with the MSMS annotation strategy allowed us to structurally confirm some typical compounds, such as the flavonoids quercetin 3-rutinoside (rutin), kaempferol, isorhamnetin, malvidin 3-arabinoside, malvidin 3-sophoroside 5-glucoside, pelargonidin 3,5-di-*O*-glucoside, cyanidin 3-*O*-(6″-succinyl-glucoside), and peonidin 3-*O*-(6″-*p*-coumaroyl-glucoside), followed by some phenolic acids namely gallic acid, homogentisic acid, caffeic acid, vanillic acid, *p*-coumaric acid, and ferulic acid. Our findings are in accordance with existing literature on this functional plant matrix, showing that elderberries are a source of bioactive compounds mainly belonging to flavonoids and phenolic acids. Domínguez et al. [9] showed that the most abundant compound of the elderberry extract was rutin (813.08 µg/100 g dry weight), followed by quercetin (228.83 µg/100 g dry weight). Similar findings were previously reported by Thomas et al. [22], confirming rutin as the most abundant phenolic compound in nine different elderberry genotypes (about 138 mg/kg). The phenolic profile recorded for the elderberry extracts is extremely interesting due to the ability of these compounds to prevent free radicals’ formation and propagation because of their high reactivity with biological molecules, such as lipids, proteins, DNA, etc., leading cell and/or DNA oxidative damage [23]. However, as well-known, the bioactive composition of elderberry is dependent on different factors, such as cultivar, location, ripening stage, and climatic conditions [24]. Therefore, different species and growth conditions of the plant may have determined the differences observed [24]. Regarding the anthocyanins composition of the elderberry extract under investigation, the untargeted annotation workflow (UHPLC-HRMS) allowed us to extend the results previously reported by our research group [9]. In this regard, we found 79 compounds classified under the sub-class anthocyanins, with cyanidin 3-gentiobioside being the most abundant compound (i.e., 7.8 mg/100 g), followed by delphinidin and glycosidic forms of malvidin and pelargonidin (Appendix A). Additionally, as reported in Appendix A, the semi-quantitative analysis based on UHPLC-HRMS allowed us to detect a total content in the raw extract equal to 73.6 mg/kg of anthocyanins (expressed as cyanidin equivalents).According to the literature, elderberry fruits are characterized by a high content of anthocyanins, which give them their characteristic dark-purple color [25]. Anthocyanins are widely used as natural pigments in the food industry [10] and, as well as other flavonoids (e.g., quercetin derivatives), exhibit antioxidant, anticarcinogenic, immune-stimulating, antibacterial, antiallergic, and antiviral properties. Besides, anthocyanins have gained great interest as functional compounds in food colorants and as potent agents against oxidative stress, reducing oxidative damage to the human body. Regarding other phytochemical classes described for elderberry [10], several antioxidant compounds belonging to tocotrienols were structurally annotated, such as α-tocotrienol, 11′-Carboxy-γ-tocotrienol, 9′-Carboxy-γ-tocotrienol, d-Tocotrienol, and d-tocopherols. In the last years, several research works have been carried out seeking the use of extracts of spices, fruits, and vegetable residues as antioxidants in meat products [6,10]. Taken together, our findings on the phytochemical profile of elderberry here investigated, combined with a strong in vitro antioxidant potential [8], highlight this extract as a potential and valuable extender to preserve meat from protein and lipid oxidation during the shelf-life period.

### 3.2. Multivariate Statistical Discrimination of Beef Burgers during Storage

In the last years, foodomics-based approaches have emerged as a valuable tool to assess the overall quality and safety of animal products [16,17,26]. Therefore, in this work, an untargeted metabolomics approach based on UHPLC-HRMS was used to depict the chemical changes of beef burgers during the shelf-life period, mainly when considering the comprehensive modifications induced by lipid and protein oxidation phenomena. The untargeted UHPLC-HRMS approach combined with the most comprehensive database on food constituents, i.e., FoodDB, revealed the presence of 470 metabolites, mainly belonging to the chemical classes of amino acids, peptides, benzenoids, fatty acyls, glycerophospholipids, medium-chain aldehydes, keto acids, polyphenols, and prenol lipids. A detailed list containing all the meat metabolites annotated according to a Level 2 of confidence against the FoodDB, together with the composite mass spectra (i.e., MS1 isotopic spectrum and MS/MS spectrum), is reported as Appendix A.

The high complexity of the metabolomic dataset in terms of the distribution of different chemical classes (such as lipids and peptides) is quite indicative of the matrix analyzed. This might be of great concern, considering that meat components interact during storage. Therefore, because of the high amounts of oxidation catalysts (such as iron and myoglobin) in meats, these components are definitely vulnerable to oxidative processes [27]. Accordingly, the main factors driving lipid oxidation in meat are represented by the fat content and fatty acid composition because fatty acids are the substrate of oxidation processes. In meat, lipids are organized into triglycerides and phospholipids, with low contributions of other types of lipids, such as free fatty acids and cholesterol [2]. However, in the last years, great attention has been devoted to the presence and role of phospholipids as reactive compounds and pivotal in the development of lipid oxidation. Therefore, by inspecting the metabolomic dataset (Appendix A), we found a good representation of phospholipids and other compounds involved in both lipids and protein oxidation processes (such as aldehydes, ketones, and other carbonyl compounds). For example, among the secondary products of lipid oxidation, we found several aldehydes and ketones, such as 4-hydroxy-2-nonenal, (E)-5-nonen-2-one, 2,4-heptadienal, 2,4,7-decatrienal, and 3,6-undecadienal (Appendix A). The abundance of these aldehydes usually presents a strong correlation with TBARs and sensory scores, thus confirming the great utility of these compounds when used as markers of lipid oxidation [28]. Interestingly, the metabolomic dataset also showed the presence of some organonitrogen compounds, such as spermine and its precursor sperminidine. A similar distribution of polyamines was also noticed in a previous work published by our research group [16] dealing with the shelf-life of packaged pork burgers added with a pitanga leaf extract.

Because of the high number of metabolites annotated and related to both oxidation processes and potential preserving effects of the prepared elderberry extract, multivariate statistics were used to facilitate the data interpretation and management. Firstly, an unsupervised hierarchical cluster analysis (HCA) from the fold-change-based heat map on the annotated metabolites was inspected and reported in Figure 2.

As clearly highlighted by the HCA heat map, two main clusters could be observed: on one side, we found all burger samples at T0, while on the other side, it represented the impact of storage time, consisting of burger samples at T5 and T13 days. The heat map showed a certain shift in metabolomic signatures during storage, also suggesting important differences between elderberry-added burgers and the other two sample groups (i.e., CON and ERY). Accordingly, CON and ERY burgers were included in the same cluster at both 5 and 13 days of storage, thus suggesting a potentially great impact of both elderberry concentrations (i.e., EE 2.5 and EE 5) on the chemical profile of beef burgers. Therefore, by looking at the unsupervised statistical findings, we were able to confirm a great impact of elderberry extracts added to beef burgers during the entire storage period, and this impact was completely different when compared with the chemical modulations exerted by the synthetic antioxidant sodium erythorbate (ERY).

As the next step, to confirm the trends highlighted by the unsupervised HCA, a supervised method based on OPLS regression (OPLS-DA) was applied to the metabolomic dataset. In particular, the OPLS-DA score plot was built considering as class membership criterion a combination of both storage time (i.e., T0, T5, and T13) and type of burgers analyzed (i.e., CON, EE 2.5, EE 5, and ERY). The resulting output is provided in Figure 3.

The supervised OPLS-DA prediction model allowed us to confirm the results observed in the HCA heat map. In this regard, the score plot outlined clear discrimination as a function of the storage time, while the chemical differences induced by the addition of elderberry extracts were particularly evident at T13 of the storage period (Figure 3), and this was true for both concentrations tested (i.e., EE 2.5 and EE 5). Besides, the supervised approach suggested that the antioxidant effect of ERY was lower during the whole storage period when compared with both concentrations of the encapsulated elderberry extract. The high prediction ability of the OPLS-DA model built was confirmed by inspecting its goodness parameters; in particular, we found an R^2^Y(cum) (goodness of fitting) = 0.858 while a Q^2^(cum) (goodness of prediction) = 0.60, with an adequate statistical significance (*p*-value of cross-validation ANOVA < 0.01). Besides, no outliers were found by inspecting Hotelling’s T-squared distribution (not shown), while the permutation testing (N = 100 random permutations) excluded overfitting phenomena. Taken together, the results of both unsupervised and supervised statistics suggested a marked impact of the encapsulated elderberry extracts on meat composition during the storage period. Therefore, in the next part of this work, we mainly evaluated the antioxidant role of elderberry as a meat extender by focusing on both classical parameters (such as MDA production resulting from lipid oxidation events and protein oxidation) and discriminant marker compounds from untargeted metabolomics.

### 3.3. Effect of Elderberry Extract on Burger Lipid and Protein Oxidation during Storage

As previously stated and widely recognized, lipid and protein oxidation represents the most important events able to affect meat quality, determining the production of volatile compounds that are potentially correlated to several off-flavors and deterioration of the product. The by-products generated by lipid oxidation can be classified into two groups [29]; the first one includes hydroperoxides and conjugated dienes (called primary end products), while the second group (including the so-called secondary end products) is characterized by other compounds such as isoprostanes, prostaglandin (PG) F2-like compounds, carbonyls (i.e., ketones and aldehydes), furans, and MDA. Therefore, in this work, we focused attention on both classical oxidation parameters (such as MDA and number carbonyl compounds) and discriminant markers provided by metabolomics. The effect of EE on the lipid and protein stability of beef burgers during 13 days of storage is shown in Table 1.

The addition of EE at all levels tested (EE 2.5 and EE 5) significantly (*p* < 0.001) inhibited both lipid and protein oxidation reactions compared to CON and ERY samples throughout storage time. Looking at five days of storage, EE 5 was particularly effective in preserving lipid oxidation, recording 0.326 mg MDA/kg, while both extracts tested (EE 2.5 and EE 5) were able to reduce protein oxidation at this time point when compared with ERY and CON samples (Table 1). Similar trends could be detected at 13 days of storage, where EE 5 was allowed to recover the lowest values for both MDA and the number of carbonyls, thus demonstrating a great ability to reduce both lipid and protein oxidation reactions over time. Despite a few differences between EE 2.5 and EE5 in terms of effectiveness against lipid oxidation, both treatments allowed to keep below 2.0 mg MDA/kg, which is accepted as a deterioration level [30]. Also, looking at the MDA produced over the shelf-life period, the CON samples recorded a significant 3.6-fold increase moving from T0 to T13, while the synthetic antioxidant sodium erythorbate determined a 15.1-fold higher increase of MDA over the shelf-life period, moving from 0.357 (T0) up to 5.393 (T13). Therefore, no significant differences in terms of lipid oxidation could be highlighted between ERY and CON samples at T13, although a significant difference could be observed at T0. Interestingly, as reported in Table 1, similar trends could be detected by looking at the protein oxidation, with no significant differences recorded between ERY and CON samples following 13 days of the shelf-life period, although a significant difference could be observed at T0.

Overall, as a general consideration, the active compounds of the extract were likely responsible for better lipid oxidation inhibition when compared with the synthetic antioxidant tested. These findings are corroborated by previous works dealing with meat extenders and active extracts, stating that natural extracts had more antioxidant power in real meat products than synthetic antioxidants, thus suggesting the possibility of using these extracts as replacers of commercial additives [6]. Another important observation of these findings is that the utilization of the lowest concentration of EE (2.5 g/kg) would be sufficient to improve the oxidative profile of both CON and ERY beef burgers. Also, the protective effect of EE against the oxidation process is related to the high amounts of phenolic compounds characterizing this natural extract. Polyphenols can inhibit or delay the oxidation of red meat mainly through chelating transition metal ions (iron of myoglobin), scavenging free radicals, and carbonyl compounds to inhibit lipid oxidation [31]. These compounds quickly donate a hydrogen atom or electrons to lipid radicals as a way to block lipid oxidation progress [31]. Also, their radical intermediates are relatively stable due to delocalization and lack of suitable sites; therefore, they can prevent lipid oxidation by blocking the chain reaction of free radicals [31]. Just to provide some examples, the flavonol rutin and the anthocyanin cyanidin (both belonging to the flavonoids class) have already been described in scientific literature as able to scavenge free radicals, remove reactive species of oxygen, inhibit lipid peroxidation, chelate metal ions, and form stable oxidized forms [31]. Regarding protein oxidation, in the redox process, some free amine groups on lysine residues are oxidized to carbonyl groups, which subsequently react with the primary amines to form Schiff base adducts. However, according to the mechanisms previously cited, phenolic compounds can protect proteins from oxidative modifications and inhibit and/or slow down carbonyl formation [32].

Following the inspection of MDA and the number of carbonyls produced during 13-days storage, we used untargeted metabolomics to shed light on the modulation of oxidative processes imposed by EE addition to the samples. Therefore, an additional OPLS-DA model was built to evaluate more selectively the changes of meat metabolomic profiles exclusively at 13 days of storage (i.e., the time-point imposing the highest variation in Figure 4). The new OPLS-DA score plot considering the differences at 13 days of storage is provided in Figure 4.

Looking at the new OPLS-DA score plot, the orthogonal latent vector of the model was found to discriminate EE-added burgers from CON and ERY samples. It is important to highlight that the OPLS regression is able to filter out some variance in the X-matrix unrelated to Y, thereby producing results that are easier to interpret. Accordingly, the new prediction model allowed us to also observe a certain degree of discrimination between CON and ERY beef burgers (Figure 4), although both samples were included in the right part of the elliptical space. The model built was characterized by excellent goodness-related parameters, recording values of 0.99 and 0.83 for R^2^Y(cum) and Q^2^(cum), respectively. Also, the absence of overfitting and outliers in the model (not shown) demonstrated that the metabolites annotated were particularly useful in describing the global changes of meat metabolites at T13 of the storage period. A variables’ selection method based on VIP (acronym for variables’ importance in orthogonal projections) markers was exploited to explore the metabolomic perturbations induced by the addition of elderberry extracts on beef burgers during storage. This approach provides the so-called VIP score (i.e., the contribution a variable makes to the OPLS-DA model), thus outlining those metabolites better explaining the hyperspace separation observed in Figure 4. The most interesting VIP markers (as related to lipid and protein oxidation) possessing a discriminant score higher than one are reported in Table 2, considering each main chemical class with the corresponding Log Fold-Change (FC) variation against the CON sample. Additionally, among the discriminant markers, we found 39 metabolites included in the class of amino acids, peptides, and analogues, followed by 13 benzenoids, 40 fatty acyls, 33 glycerophospholipids, 6 medium-chain aldehydes, 6 organooxygen compounds, 5 polyphenols, 21 prenol lipids, and 51 other metabolites (Appendix A).

When looking at the discriminant compounds shown in Table 2, it is possible to highlight the great impact of redox impairment during the shelf-life period of beef burgers. The most represented pathway involved the oxidation of polyunsaturated fatty acids (linoleic and linolenic acid derivatives), determining the appearance of hydroxy- and epoxy-forms of fatty acyls (such as 9,10-Epoxyoctadecanoic acid, 12-Hydroxy-8,10-octadecadienoic acid, and 9-hydroperoxy-10,12-octadecadienoate, among the others). According to the literature [2], lipid oxidation is the main non-microbial cause of quality deterioration in meat and meat products, reducing both the nutritional and sensory quality of meat. The unsaturated fatty acids characterizing the meat matrix react with molecular oxygen via a free radical mechanism to produce the so-called hydroperoxides (considered the first oxidation products of the oxidation pathway). The hydroperoxides are practically odorless and do not provide a significant contribution to the aroma of the products. However, the strong instability of hydroperoxides is related to their rapid decomposition in a large number of secondary reaction products, including hydrocarbons, aldehydes, ketones, alcohols, esters, and acids, which cause the development of off-flavors and off-odors in meat. Several of these oxidation products have been successfully annotated by the untargeted UHPLC-HRMS (Appendix A), thus revealing a great ability of untargeted metabolomics coupled with different multivariate statistical approaches to study the oxidation phenomena characterizing the meat matrix. When looking at those compounds likely associated with lipid oxidation processes, aldehydes are considered the most important breakdown products and the largest contributors to volatile flavors in the meat. This aspect is related to the fact that they have a low odor threshold and that they are present in significant quantities in the products that suffered oxidation processes [2]. Accordingly, in our experimental conditions, among the most important discriminant compounds following 13 days of storage, we found three aldehydic compounds, namely 4-hydroxy-2-nonenal (VIP score: 1.05), 2,6-dimethyl-5-heptanol (VIP score: 1.06), and 2,6-octadienal (VIP score: 1.12). Beyond modifying meat’s aromatic profile, aldehydes are also important because they can react with proteins causing changes in both nutritional and organoleptic properties.

Our findings clearly demonstrated a great impact of both EE concentrations added to meat on lipid oxidation events; this was true not only by inspecting the LogFC values of the aldehydes reported previously but also when monitoring the trends of hydroperoxy-fatty acids and glutathione (Table 2). The compound 4-hydroxy-2-nonenal is the primary α,β-unsaturated hydroxyalkenal that is produced by lipid peroxidation in cells. It is found throughout animal tissues and in higher quantities during oxidative stress due to the increase in the lipid peroxidation chain reaction and following stress conditions. In detail, it is generated following the oxidation of lipids containing polyunsaturated omega-6 acyl groups, such as arachidonic or linoleic groups, and of the corresponding fatty acids, namely the hydroperoxy precursors to 15-hydroxyeicosatetraenoic acid and 13-hydroxyoctadecadienoic acid, respectively. The maximum and significant reduction of 4-hydroxy-2-nonenal was recorded following the addition of EE 5, recording a LogFC value equal to −2.18, followed by EE 2.5 (LogFC: −1.31) and ERY (LogFC: −0.04).

Regarding glutathione, it is well known that the glutathione system is the major endogenous antioxidant machinery that protects animal cells from oxidative damage. Our findings also revealed a great abundance of GSH in beef burgers added with EE (at both concentrations), recording an average LogFC value equal to 2.2 (Table 2), while the utilization of ERY was not effective in preserving the GSH system of beef meat, recording a negative LogFC value (−0.30). The LogFC values measured for glutathione demonstrate a lower redox impairment of beef burgers added with elderberry extracts during shelf-life, likely due to the antioxidant role exerted by polyphenols and other bioactives characterizing the phytochemical profile of this plant matrix (as reported in Section 3.1 and according to the reviewed scientific literature). Interestingly, we found an overall down-accumulation of cholesterol in beef burgers added with increasing levels of elderberry extracts; however, no cholesterol oxidation products annotated (such as 7beta-hydroxycholesterol) were found to significantly change during the shelf-life period under investigation. The negative LogFC values detected for cholesterol following the addition of elderberry extracts (Table 2) are in contrast with the existing scientific literature and represent an unexpected finding [2]. Accordingly, cholesterol oxidation in meat products has been described to be promoted by the oxidation of coexisting PUFA in the meat matrix [2]; however, in our experimental conditions, we found that PUFA were more protected by both EEs under investigation when compared with ERY and CON samples. Therefore, this work suggests that the phytochemical composition of EEs was particularly effective in promoting a lower oxidation degree of PUFA, but this was not true when considering the modifications of cholesterol, although no cholesterol oxidation products were characterized by significant variations during the entire shelf-life period. Overall, this trend could be a consequence of several combined reasons, such as the dilution of fat with elderberry extracts and the potential prooxidant role exerted by some phytochemicals (at certain concentration levels) such as phenolic compounds (anthocyanins) and carotenoids [33]. Therefore, taken together, the findings related to lipid oxidation phenomena (both from untargeted metabolomics and MDA production) confirmed a greater ability of EE 5 when compared with EE 2.5 and sodium erythorbate, to prevent lipid deterioration during the shelf-life of beef burgers packaged in MAP conditions (80:20 O_2_:CO_2_) using polystyrene trays, which were sealed with polyethylene film.

### 3.4. Pearson’s Correlation Coefficients

Finally, to confirm the validity of the discriminant compounds outlined by untargeted metabolomics, the MDA and total carbonyl contents produced at 13-days of shelf-life were correlated with the relative abundance of each VIP metabolite reported in (Table 2); The corresponding Pearson’s correlation coefficients are then reported in Table 3.

Overall, the MDA and carbonyl contents were significantly correlated (0.05 < *p* < 0.01) with 9 VIP discriminant compounds, mostly including aldehydes, carbonyl compounds, and hydroxy/peroxy-derivatives of fatty acids. Therefore, our findings confirmed the potential of these metabolites to act as potential markers of the oxidation phenomena under investigation. In particular, the MDA content was particularly correlated with (Z)-1,5-octadien-3-one, followed by the group of medium-Chain aldehydes (including, among the others, the marker 4-hydroxy-2-nonenal). Regarding protein oxidation, the carbonyl content established highly significant correlation coefficients with two fatty acids-derived metabolites, namely (11R,12S,13S)-Epoxy-hydroxyoctadeca-cis-9-cis-15-dien-1-oic acid and 9,10-epoxyoctadecanoic acid (Table 3). Taken together, the combination of classical spectrophotometric assays and high-resolution untargeted metabolomics allowed us to confirm that EEs could be considered an effective alternative to synthetic antioxidants (ERY) to extend the shelf-life of beef burgers, also improving its overall chemical profile. Besides, the untargeted metabolomics approach demonstrated a great potential for meat science-related applications, providing a comprehensive and holistic overview and covering both primary and secondary oxidation products in the meat matrix under investigation. As also reported in our previous work [16], a full standardization of the quantitative metabolomics-based workflows will be essential to obtain more accurate information in the future regarding the exploitation of natural extenders in meat science and as potential alternatives for meat industries.

## 4. Conclusions

In this work, an encapsulated elderberry extract (EE) (at two concentration levels) has been used as an alternative to synthetic antioxidants to evaluate its ability as meat extenders, i.e., as a tool to modulate the lipid and protein oxidation reaction during 13-days of shelf-life. The EE added to beef burgers determined several metabolomic changes, likely due to the abundance of different antioxidant compounds, such as anthocyanins, flavonols, and phenolic acids. The marked chemical changes observed in all manufactured samples (i.e., control, sodium erythorbate, and EE) indicated a strong ability of EE to modulate the oxidation processes towards the entire storage period. Accordingly, multivariate statistics of metabolomics data allowed us to highlight several metabolites related to lipid and protein oxidation phenomena. Most of the discriminant metabolites outlined by metabolomics were found to be all up-accumulated in the control and sodium erythorbate samples when compared with EEs-treated samples. In particular, the main differences could be attributed to a specific set of metabolites, namely glutathione, 4-hydroxy-2-nonenal, hydroxy/peroxy-derivatives of fatty acids, carbonyl compounds (such as 5-nonen-2-one and 1,5-octadien-3-one), and cholesterol. Also, significant correlations (*p* < 0.01) were detected between malondialdehyde, total carbonyls, and these discriminant metabolites. Therefore, taken together, our findings confirmed that the combination of classical markers together with new marker compounds by untargeted metabolomics analysis represents a novel and valuable approach to assessing the ability of natural extracts (such as those tested from elderberry) to act as meat extenders during the shelf-life of beef burgers.

## 5. Patents

The technological development and the product (encapsulated elderberry extract) are protected by patent (Application number: P202230560/Date: 23 June 2022/patent pending). This patent protects: (i) the previous stages of washing, conditioning, selection, and dehydration of elderberries, (ii) the use of the hydroalcoholic solution and its conditions for the extraction of bioactive compounds from elderberries, (iii) the stabilization of the compounds extracted by encapsulation techniques (micro- or nanoencapsulation) as well as the encapsulating agents, (iv) the final product obtained (encapsulated elderberry extract), and (v) its use as an additive in the meat industry.

## Figures and Tables

**Figure 1 antioxidants-11-02130-f001:**
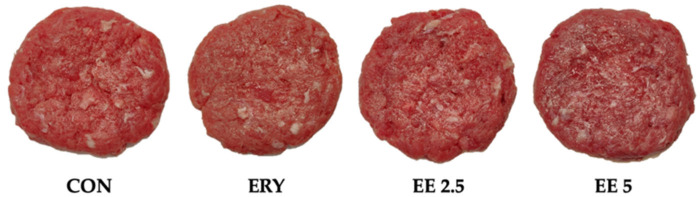
Visual aspect of the beef burgers formulated after the preparation (T0) and considering the four treatments, namely CON, ERY, EE 2.5, and EE 5.

**Figure 2 antioxidants-11-02130-f002:**
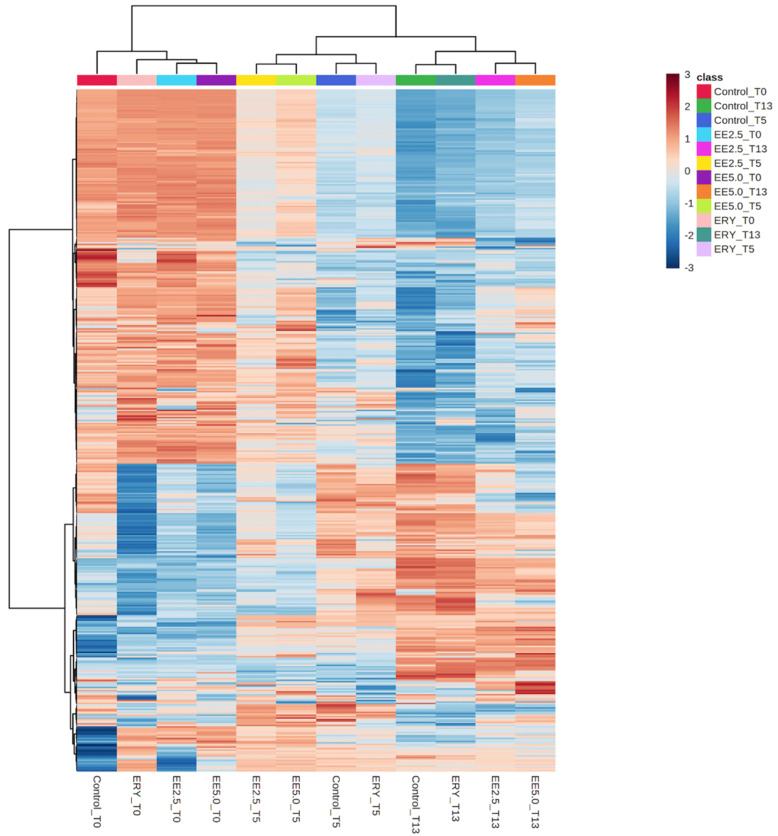
Unsupervised hierarchical cluster analysis (HCA), carried out from the fold-change distribution of each compound detected in CON, EE 2.5, EE 5, and ERY beef burgers during storage (i.e., at 1, 5, and 13 days) by using untargeted metabolomics.

**Figure 3 antioxidants-11-02130-f003:**
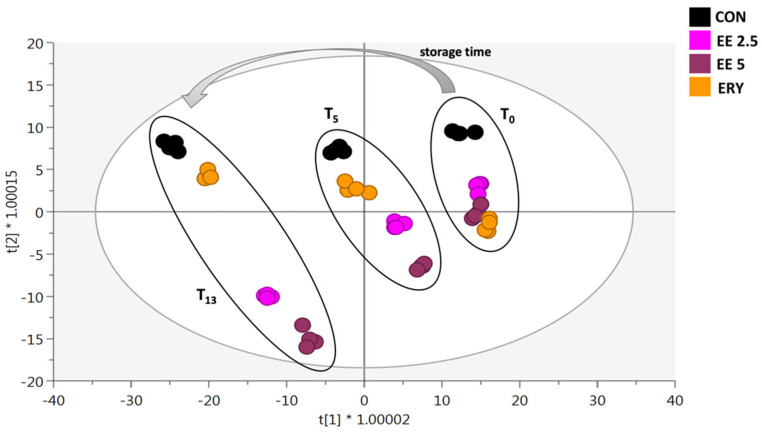
Orthogonal projections to latent structures discriminant analysis (OPLS-DA) considering the different beef burgers (CON, EE 2-5, EE 5, and ERY) during the entire storage period.

**Figure 4 antioxidants-11-02130-f004:**
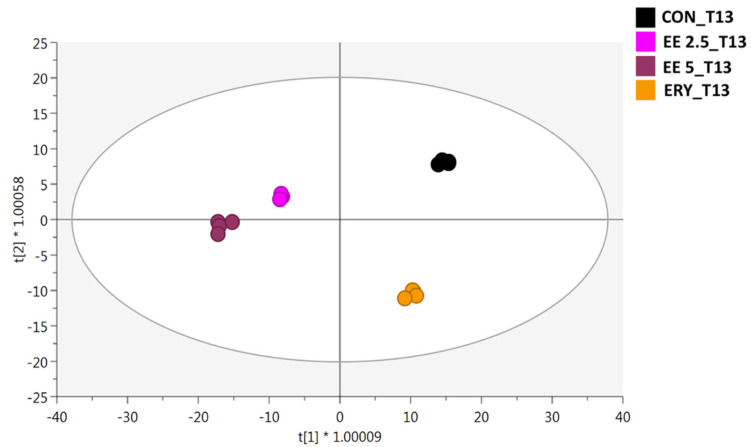
Orthogonal projections to latent structures discriminant analysis (OPLS-DA) considering the different beef burgers (CON, EE 2-5, EE 5, and ERY) at the final time-point of the storage period (13 days).

**Table 1 antioxidants-11-02130-t001:** Evolution of thiobarbituric acid reactive substances (TBARS) and carbonyl compounds in beef burgers added with EE and ERY during 1 (T0), 5 (T5), and 13 (T13) days of storage.

Treatment	MDA_T0	MDA_T5	MDA_T13	Sig.
CON	1.328 ± 0.360 ^c,A^	2.354 ± 0.817 ^c,B^	4.868 ± 0.911 ^b,C^	***
ERY	0.357 ± 0.140 ^b,A^	2.224 ± 1.312 ^c,B^	5.393 ± 1.412 ^b,C^	***
EE 2.5	0.391 ± 0.169 ^b,A^	1.056 ± 0.557 ^b,B^	2.570 ± 0.900 ^a,C^	***
EE 5	0.060 ± 0.039 ^a,A^	0.326 ± 0.369 ^a,A^	2.029 ± 1.133 ^a,B^	***
Sig.	***	***	***	
Treatment	Carbonyls_T0	Carbonyls_T5	Carbonyls_T13	Sig.
CON	2.097 ± 0.206 ^b,A^	2.570 ± 0.264 ^b,B^	4.692 ± 0.733 ^b,C^	***
ERY	1.684 ± 0.436 ^a,A^	2.533 ± 0.372 ^b,B^	4.354 ± 0.738 ^b,C^	***
EE 2.5	1.767 ± 0.412 ^a,A^	1.877 ± 0.265 ^a,A^	2.878 ± 0.474 ^a,B^	***
EE 5	1.693 ± 0.353 ^a,A^	1.671 ± 0.173 ^a,A^	2.397 ± 0.306 ^a,B^	***
Sig.	*	***	***	

Results are expressed as mean values ± standard deviation (n = 12), considering mg MDA/kg for TBARS assay and nmol carbonyls/mg protein for protein oxidation values. Sig. = significance. * = *p* < 0.05; *** = *p* < 0.001. Superscript letters within each column (^a,b,c^) and each row (^A,B,C^) indicate significant differences as provided by Duncan’s test (*p* < 0.05), considering the single shelf-life time-point and the entire shelf-life period, respectively.

**Table 2 antioxidants-11-02130-t002:** Key VIP marker compounds at T13 of the storage period (as related to lipid and protein oxidation events). Each VIP discriminant marker is followed by a VIP score, chemical class, and Log Fold-Change (FC) for the comparison against the control sample (CON).

Key VIP Marker	VIP Score	Class	LogFC (EE 2.5 vs. CON)	LogFC (EE 5 vs. CON)	LogFC (ERY vs. CON)
Glutathione	1.05	Amino acids, peptides, and analogues	2.29	2.16	−0.30
4-hydroxy-2-nonenal	1.05	Medium-Chain aldehydes	−1.31	−2.18	−0.04
2,6-dimethyl-5-heptenal	1.06	Medium-Chain aldehydes	−1.34	−2.25	−0.02
2,6-octadienal	1.12	Medium-Chain aldehydes	−1.32	−1.90	0.49
9,10-epoxyoctadecanoic acid	1.06	Fatty Acyls	−0.47	−0.54	−0.12
(11R,12S,13S)-Epoxy-hydroxyoctadeca-cis-9-cis-15-dien-1-oic acid	1.08	Fatty Acyls	−1.24	−1.24	−0.06
12-Hydroxy-8,10-octadecadienoic acid	1.12	Fatty Acyls	−0.40	−0.42	0.21
(9S,10E,12Z)-9-hydroperoxy-10,12-octadecadienoate	1.13	Fatty Acyls	−1.67	−2.98	0.70
PE(16:1(9Z)/22:4(7Z,10Z,13Z,16Z))	1.32	Glycerophospholipids	−0.15	1.10	0.37
(E)-5-Nonen-2-one	1.03	Carbonyl compounds	−0.33	−0.37	0.18
(Z)-1,5-octadien-3-one	1.09	Carbonyl compounds	−0.80	−1.03	0.22
Cholesterol	1.05	Steroids and derivatives	−0.89	−1.38	−0.05

**Table 3 antioxidants-11-02130-t003:** Pearson’s correlation coefficients considering malonaldehyde (MDA), total carbonyl content, and VIP discriminant metabolites related to lipid and protein oxidation reactions, as highlighted by OPLS-DA supervised model.

VIP Marker Compound	MDA (TBARS Assay)	Carbonyls (Protein Oxidation)
MDA	-	0.958 **
Carbonyls	0.958 **	-
4-hydroxy-2-nonenal	0.847 **	0.904 **
2,6-dimethyl-5-heptenal	0.849 **	0.903 **
2,6-octadienal	0.856 **	0.836 **
9,10-epoxyoctadecanoic acid	0.791 **	0.907 **
(11R,12S,13S)-Epoxy-hydroxyoctadeca-cis-9-cis-15-dien-1-oic acid	0.840 **	0.909 **
12-Hydroxy-8,10-octadecadienoic acid	0.817 **	0.819 **
(9S,10E,12Z)-9-hydroperoxy-10,12-octadecadienoate	0.737 *	0.703 *
(E)-5-Nonen-2-one	0.762 **	0.756 **
(Z)-1,5-octadien-3-one	0.865 **	0.877 **

* = *p* < 0.05; ** = *p* < 0.01.

## Data Availability

The data presented in this study are available in the article and Appendix A.

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
