# Peer review of "Elderberry (Sambucus nigra L.) Encapsulated Extracts as Meat Extenders against Lipid and Protein Oxidation during the Shelf-Life of Beef Burgers"

_antioxidants, 2022, doi:10.3390/antiox11112130_

Round 1

Reviewer 1 Report

The article presents a valuable study on the effect of Elderberry (Sambucus nigra L.) encapsulated extract on lipid and protein oxidation of beef burgers. The paper is original, since no other studies have already investigated the use of this encapsulated extract in beef burger formulations to control lipid and protein oxidation during the shelf-life. Furthermore, the integrated spectrophotometric assays and the metabolomics-based approach has never been evaluated. Overall, the manuscript is well organized and written. However, I do see some inaccuracies that should be improved:

Abstract section:

-line 22-23: I think there is a typo, the acronym of the d) formulation with 5000 mg/kg of encapsulated elderberry extract is the same to c) formulation with 2500 mg/kg (EE 2.5 and EE 2.5). Check it, please.

-line 358-359: It seems that significant differences could be highlighted between ERY and CON samples at T0. Check it, please.

-line 360-361: The same trend it could be observed for protein oxidation. Check it, please.

-line 371-373: Add some references, please.

-line 374-376: In this part too, please.          

-line 401: I think there is a mistake. Where is Figure 4? Check it, please.

-line 411: It could be the same mistake. Check it, please.

-line 476-477: Add some references, please.

-line 524: I think there is a typo “… an encapsulated elderberry extract (EE) FROM”. Check it, please!

- In the whole paper, it could be more appropriate to write the quantities of erythorbate and extract used in the same way: i.e., line 21-22: … 500…2500….5000 mg/kg; line 109: “sodium erythorbate at 500 ppm”; line 109-110: “… 2.5 g/kg [EE 2.5] and 5 g/kg [EE 5]”. Check it please.

Author Response

Reviewer 1

The article presents a valuable study on the effect of Elderberry (Sambucus nigra L.) encapsulated extract on lipid and protein oxidation of beef burgers. The paper is original, since no other studies have already investigated the use of this encapsulated extract in beef burger formulations to control lipid and protein oxidation during the shelf-life. Furthermore, the integrated spectrophotometric assays and the metabolomics-based approach has never been evaluated. Overall, the manuscript is well organized and written. However, I do see some inaccuracies that should be improved:

Authors: we would like to thank the reviewer for having appreciated this work. The manuscript has been revised according to each minor issue detected.

Abstract section:

-line 22-23: I think there is a typo, the acronym of the d) formulation with 5000 mg/kg of encapsulated elderberry extract is the same to c) formulation with 2500 mg/kg (EE 2.5 and EE 2.5). Check it, please.

Authors: Yes, it was a typo. Revised, accordingly.

-line 358-359: It seems that significant differences could be highlighted between ERY and CON samples at T0. Check it, please.

Authors: Indeed, following the shelf-life period of the different burgers, the CON and ERY samples showed at T13 not significant differences in terms of lipid oxidation. On the other hand, at T0 (after the formulation), CON samples resulted as more oxidated when compared with ERY, EE2-5 and EE5. We modified the sentence, accordingly.

-line 360-361: The same trend it could be observed for protein oxidation. Check it, please.

Authors: Indeed, following the shelf-life period of the different burgers, the CON and ERY samples showed at T13 not significant differences in terms of protein oxidation. On the other hand, at T0 (after the formulation), CON samples resulted as more oxidated when compared with ERY, EE2-5 and EE5. We modified the sentence, accordingly.

-line 371-373: Add some references, please.

Authors: done.

-line 374-376: In this part too, please.          

Authors: done.

-line 401: I think there is a mistake. Where is Figure 4? Check it, please.

Authors: We apologize for the mistake. It was referred to Figure 3.

-line 411: It could be the same mistake. Check it, please.

Authors: We apologize for the mistake. It was referred to Figure 3.

-line 476-477: Add some references, please.

Authors: Added, accordingly.

-line 524: I think there is a typo “… an encapsulated elderberry extract (EE) FROM”. Check it, please!

Authors: Yes, it was a typo. Revised, accordingly.

- In the whole paper, it could be more appropriate to write the quantities of erythorbate and extract used in the same way: i.e., line 21-22: … 500…2500….5000 mg/kg; line 109: “sodium erythorbate at 500 ppm”; line 109-110: “… 2.5 g/kg [EE 2.5] and 5 g/kg [EE 5]”. Check it please.

Authors: The units have been revised throughout the manuscript. Thank you for pointing it out.

Reviewer 2 Report

The manuscript investigated the effects of elderberry extracts on meat shelf life by determining metabolites and oxidative parameters, which provide a potential method to extend the meat shelf life. However, some details need further improvement.

Line 23 Replace the “EE2.5” by “EE5.0”.

Line 87 Although the product is protected, the content of main functional ingredient such as water-soluble anthocyanins should be provided.

Line 172 The analysis method of lipid and protein oxidation parameters data must supplement in this part.

Line 216-217 the manuscript mentioned “the bioactive composition of elderberry is dependent on different factors, such as cultivar, location, ripening stage, and climatic conditions”, so can you ensure the stability of elderberry extracts.

Line 361-363 The elderberry extracts supplementation levels are 5 or 10 times than sodium erythorbate, so how do you make the conclusion that natural extracts are more effective than synthetic?

Line 523 The conclusion is too long and needs to be simplified

One more question, how about the changes of meat quality such as nutrients content with the extension of shelf life?

Author Response

Reviewer 2

The manuscript investigated the effects of elderberry extracts on meat shelf life by determining metabolites and oxidative parameters, which provide a potential method to extend the meat shelf life. However, some details need further improvement.

Authors: we would like to thank the reviewer for having appreciated this work. The manuscript has been revised according to each minor issue detected.

Line 23 Replace the “EE2.5” by “EE5.0”.

Authors: it was a typo. Revised, accordingly.

Line 87 Although the product is protected, the content of main functional ingredient such as water-soluble anthocyanins should be provided.

Authors: We determined by UHPLC-HRMS (using a semi-quantitative analysis) the content of total anthocyanins in the elderberry extract. We have added a sentence in the revised version of the manuscript about this. Thank you for pointing it out.

Line 172 The analysis method of lipid and protein oxidation parameters data must supplement in this part.

Authors: We have improved this section, accordingly.

Line 216-217 the manuscript mentioned “the bioactive composition of elderberry is dependent on different factors, such as cultivar, location, ripening stage, and climatic conditions”, so can you ensure the stability of elderberry extracts.

Authors: Thank you for your comment. The enormous influence of climatic and edaphic conditions on the variations of the bioactive compounds of elderberries (as in other types of berries) is a proven fact. However, in this study, the elderberries were harvested at different times (within the same year, at optimum ripeness), in the same geographic area, and under identical climatic conditions. Therefore, the variation for elderberries was limited. In addition, despite carrying out different extractions and encapsulations (on different dates and from the starting material described above), it was found that the variation in antioxidant capacity was practically insignificant (data from another study in preparation). Therefore, we can affirm that the stability of the encapsulated elderberry extracts used, despite being made at different times, was the same.

Line 361-363 The elderberry extracts supplementation levels are 5 or 10 times than sodium erythorbate, so how do you make the conclusion that natural extracts are more effective than synthetic?

Authors: The antioxidant capacity as well as the content of TPC was previously calculated before carrying out this study. The data are part of a parallel study (on the characterization of bioactive compounds and their antioxidant capacity), and after verifying that erythorbate had approximately 10 times more antioxidant capacity than the encapsulated extract of elderberries, we decided to use these concentrations (2.5 and 5 %) (5 and 10 times more). For this reason, we can affirm that even with a total antioxidant capacity lower than that of erythorbate, it exerts a protective power against degrading oxidation processes that is greater than that of the synthetic additive.

Line 523 The conclusion is too long and needs to be simplified

Authors: the conclusions section has been shortened, accordingly.

One more question, how about the changes of meat quality such as nutrients content with the extension of shelf life?

Authors: Indeed, this preliminary work was designed to assess specifically the metabolomic changes as related to protein and lipid oxidation in beef burgers prepared with a natural antioxidant extract from elderberry, to confirm its suitability as meat extender during the shelf-life of the meat product. Therefore, considering the aim of the special issue of the journal Antioxidants (dealing mainly with meat extenders), we focused the attention on the oxidation-related processes, and we have already designed ad-hoc studies to evaluate the overall changes of meat quality (such as nutrients content), but we plan to use these data in future publications.

Reviewer 3 Report

antioxidants-1991123

The manuscript is well-prepared and the research is described appropriately. I recommend minor revisions before publication as follows:

The first sentence in the abstract should be rewritten to be more descriptive.

The authors need to provide some pictures from the study

Information about values, standard deviations, statistics etc. should be below the tables

Table 1 two dots at the end of the table caption

The first sentence in the conclusions needs to be revised since it looks unclear (beginning „from …)

Author Response

Reviewer 3

The manuscript is well-prepared and the research is described appropriately. I recommend minor revisions before publication as follows:

Authors: we would like to thank the reviewer for having appreciated this work. The manuscript has been revised according to each minor issue detected.

The first sentence in the abstract should be rewritten to be more descriptive.

Authors: the first sentence of the Abstract section has been revised and simplified, accordingly.

The authors need to provide some pictures from the study

Authors: as suggested by the reviewer, we have now added a new Figure 1, showing the visual aspect of the different burgers formulated.

Information about values, standard deviations, statistics etc. should be below the tables

Authors: revised, accordingly.

Table 1 two dots at the end of the table caption

Authors: revised, accordingly.

The first sentence in the conclusions needs to be revised since it looks unclear (beginning „from …)

Authors: the first sentence of the Conclusions section has been revised, accordingly.